# Analysis of the Association between Internal and External Training Load Indicators in Elite Soccer; Multiple Regression Study

**DOI:** 10.3390/sports10090135

**Published:** 2022-09-06

**Authors:** Sime Versic, Toni Modric, Borko Katanic, Mario Jelicic, Damir Sekulic

**Affiliations:** 1Faculty of Kinesiology, University of Split, 21000 Split, Croatia; 2HNK Hajduk Split, 21000 Split, Croatia; 3Faculty for Sport and Physical Education, University of Montenegro, 81400 Niksic, Montenegro

**Keywords:** sRPE, running performance, association, football, load monitoring

## Abstract

The aim of this study was to identify the external training load (ETL) variables that are most influential on the session rating of perceived exertion (sRPE) during elite soccer training. The participants (*n* = 29) were adult male soccer players from a single team that competed in Croatia’s highest national soccer competition in the 2021/2022 season. Data were collected using the 10 Hz Global Positioning System from 66 training sessions, and a total of 1061 training observations were undertaken. The univariate and multivariate relationships among the predictors (ETL variables) and the criterion (sRPE) were assessed using forward stepwise multiple regressions and Pearson’s correlations, respectively. ETL variables explained 63% of the variance in the sRPE (Multiple R = 0.79; *p* < 0.01), and the model was successfully cross-validated. The significant partial regressors were total distance (β = 0.66), metres per minute (β = −0.47), high-intensity accelerations (β = 0.22) and decelerations (β = 0.18), and sprint distance (β = 0.14). All ETL variables were significantly correlated with the sRPE (all *p* < 0.01), with the highest correlations found for total distance covered (*r* = 0.70) and high-intensity accelerations and decelerations (*r* = 0.62 and 0.65, respectively). Such results show that (i) the total distance and acceleration rates during the training sessions are the most important predictors of the sRPE, and (ii) a combination of different ETL variables predicts the sRPE better than any individual parameter alone. This study shows that both the volume and intensity of training are related to players’ internal responses. The findings ultimately provide further evidence to support the use of sRPE as a global measure of training load in soccer players.

## 1. Introduction

The intensity level of soccer matches has increased tremendously over the last decade [1]. To successfully cope with such increased match loads, the physical conditioning of the players has become an indispensable component of soccer training programs [2]. The stress placed on players during physical conditioning programs necessitates continuous monitoring of the training load. In general, an accumulation of stress and subsequent fatigue without adequate recovery may predispose the players to a maladaptive training response [3]. Therefore, proper training load monitoring may help in load management, which consequently may maximise physical performance, reduce occurrences of injury and illness, and minimise the risk of nonfunctional overreaching [4]. Indeed, research shows that accurate management of the training load is crucial for the planning and periodisation of training [5].

When considering training load, two components are often conceptualised according to measurable parameters occurring internally or externally to the players: (i) external training load (ETL), which presents the physical work prescribed in the training plan, and (ii) internal training load (ITL), which presents the psychophysiological responses of the individual to the ETL [6,7]. The standard methodology to monitor ETL in soccer is the quantification of running performance, such as by measuring the total distance covered, distances covered in different speed zones, accelerations, and decelerations using the Global Positioning System (GPS) [8]. On the other hand, methods based on heart rate analysis are used to monitor the ITL in soccer [9]. However, given that the use of heart-rate-based methods is not always feasible [10], subjective and self-reporting measures have become common, low-cost alternatives for monitoring ITL [11,12]. Thus, the session rating of perceived exertion (sRPE) is increasingly used as a simple, noninvasive technique for monitoring ITL [10,13,14,15]. Previously, it was shown that sRPE is highly correlated with heart rate and lactate concentrations during intermittent team sports such as soccer [4,10,16,17,18,19,20,21]. These findings confirm the validity and reliability of this method, leading to the widespread use of sRPE for monitoring ITL [3,9,16,22]. 

Although the measurement of ETL merely describes the activity a player has completed and may not accurately depict the physiological stress imposed on individual athletes [16], it appears that a growing number of coaches and scientists tend to focus on studying the ETL without considering the ITL [7,23]. Namely, acute and chronic changes in the training outcomes are ultimately the result of an athlete’s cumulative ITL over a given time period [14,24,25], which therefore places great importance on the measurement of ITL and its influential factors [26]. Moreover, similar ETL can result in different ITLs for different players due to differences in their individual characteristics (e.g., training history and actual physical fitness) [27]. Therefore, it was concluded that a combination of ITL and ETL when monitoring training loads is more appropriate as it can improve load management and help to optimise physical fitness and support injury prevention [28].

In general, the relationships between ETL and ITL have been an object of research in recent times. However, the research to date has mostly examined relationships between ETL and ITL by analysing youth or semiprofessional players [3,4,21,29], while information on the relationship between ETL and ITL among adult elite soccer players is scarce [16,30]. To the best of our knowledge, only a few studies have investigated this issue to date [27,31], while only one considered the sRPE as a measure of the ITL [25]. Specifically, Radzimiński et al. found that the total distance, high-speed running, sprinting, and the number of accelerations and decelerations were moderately related to the RPE [31], while Jaspers et al. highlighted decelerations as an important external load indicator affecting the RPE [27]. On the other hand, Gaudino et al. reported that the external load measures that were moderately predictive of the sRPE in soccer training were high-speed running distance and the number of impacts and accelerations [25]. 

Evidently, knowledge of the relationship between the sRPE and ETL among elite adult soccer players is limited; therefore, more research is needed. The information gained from studying the possible relationship between the sRPE and ETL may have a great practical impact on the development of scientific coaching of elite soccer players [21]. Specifically, a better understanding of the relationship between the sRPE and ETL could help coaches in enhancing training requirements and athlete monitoring [4]. Therefore, the aim of this study was to identify the ETL variables that are most influential on the sRPE during elite soccer training.

## 2. Materials and Methods

### 2.1. Participants and Design

The participants (*n* = 29) in this study were adult male soccer players (M ± SD, age: 26.32 ± 4.74, body mass: 77.68 ± 6.78, height: 180.95 ± 4.48, body fat: 10.75 ± 3.02) from a single team that competed in Croatia’s highest national soccer competition in the 2021/2022 season. Goalkeepers were excluded from the study due to the different physical demands of their position. All the players were informed about the research procedures, requirements, benefits, and risks, and their written consent was obtained before the study began. The study was conducted according to the requirements of the Declaration of Helsinki and was approved by the Faculty of Kinesiology, University of Split (approval number: 2181-205-02-05-19-0020).

Players were observed over all training sessions (*n* = 66) in one competitive half-season. Considering that this study focused on the relationship between the ETL and sRPE in training sessions, data from matches, on-field recovery sessions, and rehabilitation sessions were excluded from the analysis [27]. As a result, 1061 match performances were retrieved and used as cases for this study. Descriptive parameters for the ETL and sRPE data are presented in Table 1.

### 2.2. Procedures

For each training session, the ETL was measured using 10 Hz GPS and 100 Hz accelerometer technology (Vector S7, Catapult Sports Ltd., Melbourne, Australia). This system was previously investigated for its metrics and was found to be appropriately reliable and valid in sports settings (i.e., less than 1% measurement error and 80% of common variance with running speed measured by timing gates) [32,33]. GPS devices were turned on 15 min before training sessions, and each player wore the same GPS device in all training sessions in order to avoid inter-unit variability [2]. The ETL variables included: total distance covered (m); metres per minute (m/min); running zone (4–5.5 m/s) distance (m) and efforts (count), high-speed running zone (5.5–7 m/s) distance (m) and efforts (count); sprinting zone (>7 m/s) distance (m) and efforts (count); high-intensity running zone (>5.5 m/s) distance (m) and efforts (count); high-intensity accelerations (>3 m/s^2^); and high-intensity decelerations (less than −3 m/s^2^) [34].

Each player’s rating of perceived exertion (RPE) was collected in isolation, 20 min after each training session, using the curvilinear CR-10 Borg scale modified by Foster et al. [14]. The RPE was derived by asking each player, “How hard was your session?” with 1 being very, very easy and 10 being maximal exertion. All the players were fully familiarised with the use of the scale. The sRPE was subsequently calculated by multiplying the training duration (in minutes) by the RPE, as described by Foster et al. [4,14,27]. 

### 2.3. Statistical Analyses 

Normality was confirmed using the Kolmogorov–Smirnov test, and homoscedasticity was proven by the Levene test. The univariate relationships between the sRPE and ETL variables were calculated using Pearson’s correlation coefficients, with the coefficient classification as previously suggested: *r* ≤ 0.35 indicates a low or weak correlation, *r* = 0.36 to 0.67 indicates a modest or moderate correlation, *r* = 0.68 to 1.0 indicates a strong or high correlation, and *r* > 0.90 indicates a very high correlation [35]. 

The multivariate relationships among the predictors and the criterion (sRPE) were assessed via forwards stepwise multiple regression analysis. In the first phase, multiple regression was calculated using half of the observations (*n* = 531; the randomly selected validation sample). The regression model equations were then applied to the remaining half of the observations (*n* = 530; the cross-validation sample). The actual performance scores of the cross-validation sample were then correlated to their predicted (calculated) performance scores. Afterward, the calculated and achieved performance scores were compared by means of a t-test for dependent samples. Finally, the model was validated with a Bland–Altman plot of the average calculated and achieved scores (abscissa) and the differences between the achieved and calculated scores (ordinate) with regard to 1.96 SD of difference [36]. All statistical analyses were conducted using Statistica v.15 (Statsoft, Tulsa, OK, USA), and *p* < 0.05 was considered to indicate statistical significance.

## 3. Results

Significant correlations were found between the sRPE and all ETL variables (all *p* < 0.01). In detail, total distance was found to be highly correlated with the sRPE (*r* = 0.70), while metres per minute, running distance, running efforts, high-speed running distance, high-speed running efforts, high-intensity running distance, high-intensity running efforts, high-intensity accelerations, and high-intensity decelerations were moderately correlated with the sRPE (*r* = from 0.38 to 0.65). The lowest correlations with the sRPE were found for sprint distance and sprint efforts (*r* = 0.28 and 0.30, respectively) (Table 2).

When multiple regression was calculated for the sRPE in the validation subsample of the participants, the predictors explained 63% of the variance in the criterion. The significant partial regressors were total distance (β = 0.66, *p* = 0.01), metres per minute (β = −0.47, *p* = 0.01), high-intensity accelerations (β = 0.22, *p* = 0.01), sprint distance (β = 0.14, *p* = 0.01), and high-intensity decelerations (β = 0.18, *p* = 0.01) (Table 3).

The sRPE regression model obtained for the validation subsamples was as follows: 

sRPE = 140.56 + 0.08 × “Total distance”−4.60 × “Meters per minute” + 1.15 × “High-intensity accelerations” + 0.41 × “Sprint distance” + 1.00 × “High-intensity decelerations” + 0.06 × “Running distance”−0.09 × “High-speed running distance”.

When the regression model was applied to the cross-validation subsample, the common variance between the calculated and observed scores was 56% (Pearson’s *r* value of 0.75 between the calculated and observed scores; *p* < 0.05). As a result, the validity of the regression model was defined as appropriate (the common variance between the observed and calculated scores was greater than 50%). In the next phase, the calculated and observed scores for the sRPE were compared by means of a T-test for dependent samples. There was no significant difference between the calculated and observed scores for the cross-validation subsample (322.45 ± 125.02 and 318.43 ± 97.35, respectively; t value = 1.63, *p* = 0.10).

The Bland–Altman plot showed that almost all the cross-validating scores were positioned between ±1.96 SD in the sRPE score differences (observed minus predicted scores) (Figure 1). This confirmed that the regression modelling for the sRPE was appropriate.

## 4. Discussion

This study aimed to identify the ETL variables that are most influential on the sRPE during elite soccer training. The key finding was that (i) the total distance and acceleration rates during the training sessions were the most important predictors of the sRPE, and (ii) a combination of different ETL variables predicted the sRPE better than any individual parameter alone. The findings further demonstrate that the sRPE can be a useful tool for monitoring the training load among elite soccer players. 

In recent years, researchers have become increasingly interested in the relationship between ITL and ETL measures in soccer players, with most of these studies assessing the ITL by way of the RPE among subelite or young players [3,4,21,27,29,31]. Surprisingly, although recent research has shown that the sRPE may be a more useful tool for monitoring training load since the RPE does not reflect the intensity of a training session [4], the relationship between the sRPE and ETL has rarely been examined. In general, the relationship between the sRPE and ETL has been comprehensively investigated in elite rugby league and Australian football players [16,30], while only one study investigated this issue among elite adult soccer players [25]. Thus, Gaudino et al., in their study, aimed to determine the ETL variables that are most influential on the sRPE in elite soccer players, and they demonstrated that a combination of ETL variables predicts the sRPE better than any individual parameter alone. More specifically, considering that high-speed running and the numbers of impacts and accelerations were found to best predict the sRPE during elite soccer training, the authors concluded that a combination of speed, acceleration, and impacts is likely to be a strong predictor of the sRPE in soccer [25].

The findings from our study, in general, support such conclusions, but some differences can be observed. Namely, the results of multiple regression modelling in our study confirm that a combination of different ETL variables may predict the sRPE better than any individual parameter alone. In particular, a combination of the herein selected predictors was more strongly correlated with the sRPE (Multiple R = 0.79; *p* < 0.01) than any of the predictors alone. However, the most influential variables on the sRPE in our study differ from those in the study by Gaudino et al. [25]. Specifically, we found that the total distance (β = 0.66), metres per minute (β = −0.47), high-intensity accelerations (β = 0.22) and decelerations (β = 0.18), and sprint distance (β = 0.14) were the most significant partial regressors. Although these findings are in line with the study by Gaudino et al. [25], as they confirm that training intensity (i.e., defined by high-intensity accelerations/decelerations, metres per min, and distance covered at high speed) has an important role in defining the sRPE, our results additionally indicate that the volume of training (i.e., defined by the total distance covered) greatly influences the sRPE of adult elite soccer players, which was not found by Gaudino et al. [25].

A possible explanation for these differences may be found in the different training approaches of the observed players. Specifically, Gaudino et al. investigated English Premier League players, who often play 2–3 matches per week [37,38]. Consequently, players’ training sessions between matches typically focus on recovery from high-intensity match loads by utilising passing drills, shooting practice, or small-sided games [2,5,39]. In general, such training programs do not provoke a high total distance. In contrast, the current study analysed players from the Croatian first league, who mostly play one match per week. Given that the training programs of those players are focused not only on recovery but also on improving players’ physical capacities, their training sessions provoked a higher total distance. These considerations can directly be supported by analysing the average session total distance in both studies. In particular, players from the current study covered a ~20% higher total distance during the training sessions when compared to the players from the study by Gaudino et al. (4269 m and 3545 m, respectively) [25]. Therefore, their ratings of perceived exertion were most likely additionally defined by the session’s total distance, explaining why the training volume of the herein-analysed players greatly influenced the sRPE. 

The results of regression modelling suggesting that both the intensity and volume of training are important aspects of players’ internal responses can be directly supported by analysing the univariate correlations between the predictors and criterion. Specifically, all variables that define training volume (e.g., total distance covered) and intensity (e.g., metres per minute; high-speed, sprint, and high-intensity running distances and efforts; and high-intensity accelerations and decelerations) were significantly correlated with the sRPE (all *p* < 0.01). However, it must be emphasised that those variables that define training intensity were differently correlated with the sRPE. Most specifically, higher correlations were found for high-intensity accelerations and decelerations (*r* = 0.62 to 0.65) than for covered distances (*r* = 0.28 to 0.45) and number of efforts (*r* = 0.30 to 0.46) at higher speeds. This indicates that players’ internal responses during the training session were most strongly influenced by higher acceleration rates. Such results are in line with the study by Gaudino et al., which highlighted acceleration rates as one of the most influential variables on the sRPE [25]. 

Considering that significant emphasis has been placed on distances covered at higher speeds when examining the physical demands in soccer [40,41], it is noteworthy that our findings additionally underline the importance of accelerations in soccer. Namely, soccer involves a number of acyclical changes in activity, each characterised by accelerations that further increase the energy demands placed on the athlete, even when running within a low-speed threshold [25,42,43]. Therefore, acceleration rates are of significant importance when examining the overall physical demands of team sports such as soccer [25,42,43]. It is also noteworthy that the findings from the current study further emphasise the importance of examining the total distance during a training session. This is particularly important to highlight since it was repeatedly reported that the total distance covered is a non-essential variable when examining physical demands in soccer [44,45]. However, given our findings, which suggest that the total distance covered most highly influences the players’ internal response during the training session, examining the overall physical demands during the soccer training programs of adult elite soccer should indispensably include values of the overall covered distance.

The main limitations of this study are associated with the analysis of the relationship between the ETL and ITL for all players together, and further research should focus on individual dose–response relationships [4]. In addition, the findings of the current study were derived from entire training sessions. Since different parts of training elicit different physical demands, future studies should evaluate the relationship between ETL and ITL across a range of soccer-specific training drills to derive a deeper understanding of the degree to which different factors affect the sRPE [25]. Moreover, the herein-analysed players were members of a single team, which may have had a significant impact on the results. However, this is a very common obstacle in studies involving professional and elite players [46,47]. Finally, in this study, we observed normally distributed variables only, and consequently, parametric statistical methods were applied. Moreover, some potentially important non-parametric variables were not used in the study, which would allow additional insight into the studied relationships. 

## 5. Conclusions

This study provides important information on the understanding of the relationship between the ETL and ITL during adult elite soccer training programs. Specifically, determining which ETL variables have the greatest influence on the perception of effort enables coaches to better monitor athletes and enhance training prescription; as a consequence, they can both reduce the risk of injury and improve physical performance [4]. The findings show that the volume of training sessions (i.e., total distance covered) has the greatest impact on the sRPE. In addition, the intensity of the training, particularly high-intensity accelerations and decelerations, was also shown to greatly impact the sRPE. These results suggest that both the volume and intensity of training are related to players’ internal responses. These findings ultimately provide further evidence to support the use of the sRPE as a global measure of training load in soccer players.

## Figures and Tables

**Figure 1 sports-10-00135-f001:**
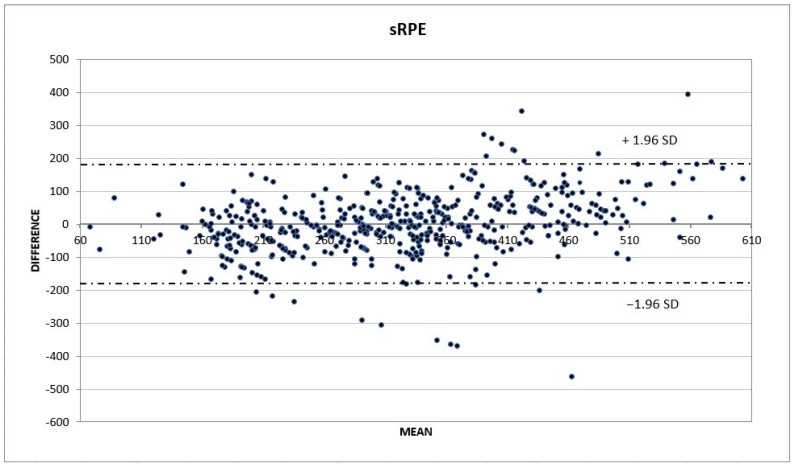
Bland Altman plot of the actual and predicted scores for sRPE.

**Table 1 sports-10-00135-t001:** Descriptive parameters for ETL variables and sRPE.

	Mean	Minimum	Maximum	SD
Total distance	4269.3	1202.0	7611.0	1044.9
Metres per minute	65.8	27.5	107.9	13.3
Running distance	365.6	0	1618.0	202.4
Running efforts	38.9	0	118.0	19.8
High-speed running distance	99.1	0	518.0	86.6
High-speed running efforts	7.5	0	35.0	6.3
Sprint distance	19.4	0	312.0	40.8
Sprint efforts	1.1	0	13.0	1.9
High-intensity running distance	118.5	0	746.0	116.9
High-intensity running efforts	8.6	0	47.0	7.6
High-intensity accelerations	58.6	2.0	141.0	24.6
High-intensity decelerations	50.3	1.0	129.0	22.6
sRPE	322.5	64.0	756.0	125.0

SD = standard deviation.

**Table 2 sports-10-00135-t002:** Pearson’s correlation coefficients between sRPE and predictor variables.

	*r*	*p*
Total distance	0.70	0.01
Metres per minute	0.38	0.01
Running distance	0.59	0.01
Running efforts	0.63	0.01
High-speed running distance	0.44	0.01
High-speed running efforts	0.46	0.01
Sprint distance	0.28	0.01
Sprint efforts	0.30	0.01
High-intensity running distance	0.42	0.01
High-intensity running efforts	0.45	0.01
High-intensity accelerations	0.65	0.01
High-intensity decelerations	0.62	0.01

*r* = Pearson’s correlation coefficient, *p* = level of significance.

**Table 3 sports-10-00135-t003:** Forward stepwise linear regression of sRPE calculated for validation sample.

	β	SE(β)	b	SE(b)	t-Value	*p*-Value
Intercept			140.56	22.12	6.35	0.01
Total distance	0.66	0.05	0.08	0.01	12.84	0.01
Metres per min	−0.47	0.04	−4.60	0.42	−11.02	0.01
High-intensity accelerations	0.22	0.07	1.15	0.38	3.06	0.01
Sprint distance	0.14	0.04	0.41	0.11	3.74	0.01
High-intensity decelerations	0.18	0.07	1.00	0.39	2.56	0.01
Running distance	0.10	0.05	0.06	0.03	1.80	0.07
High-speed running distance	−0.06	0.05	−0.09	0.06	−1.32	0.19
R	0.79					
R^2^	0.63					
*p*	0.01					

β = standardised regression coefficient; SE = standard error; b = nonstandardised regression coefficient; *p* = level of significance; R = multiple correlation; R^2^: coefficient of determination.

## Data Availability

Data will be provided to all interested parties upon reasonable request.

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
