# Peer review of "Analysis of the Association between Internal and External Training Load Indicators in Elite Soccer; Multiple Regression Study"

_sports, 2022, doi:10.3390/sports10090135_

Round 1

Reviewer 1 Report

Dear Authors,

Well done in submitting a clear and informative manuscript.

Please see below minor comments:

Abstract (and throughout): Please explain clearly the negative correlation with meters per minute.

Introduction, line 37: Is it the training load monitoring that may maximise performance or does monitoring help to inform training load management?

Line 52: I would remove the word 'now' as sRPE has been used for a while. Please insert original Foster references here.

Line 70: Please add "However, the research to date has mostly"

Line 72: Whilst the research on adult elite soccer players may be scarce, please link in relevant studies here on adult elite team sport athletes.

Table 1: Please add 2 further rows: intensity within the session and volume per session. That is, the two variables you have used to calculate sRPE.

Line 125: It would be worth clarifying that the CR-10 Borg Scale is curvilinear (i.e. 5 = hard).

Discussion, page 6, line 197: Please change from 'are' to 'were' as these reflect your direct findings of the research that has been done. That is, adding to the overall body of evidence on TL.

Line 228: No need for additional end bracket here.

Line 277-278: Can you contrast your analytical methods with non-parametric methods here?

References: Please ensure consistent in the capitalisation of journal titles.

Line 198: Similarly with 'predict', change to 'predicted'.

Author Response

Reviewer 1

Dear Authors,

Well done in submitting a clear and informative manuscript.

Please see below minor comments:

RESPONSE: Thank you very much for recognizing potential of our work. We tried to follow all your suggestions and comments. Please find our responses bellow. 

Abstract (and throughout): Please explain clearly the negative correlation with meters per minute.

RESPONSE: Thank you for your comment. We must say that we did not find negative correlation between sRPE and meters per minute. As it can be seen from table 2, Pearson’s correlation coefficients between sRPE and meters per minute is 0.38 (p<0.01). As we did not state this in abstract, we believe that you minded partial regression coefficients (β) meters per minute, which is -0.47. In general, partial regression coefficients measure the expected change in the dependent variable (sRPE) associated with a one unit change in an independent variable holding the other independent variables constant. Therefore, we are pretty much assured that it cannot be stated that there is a “clear negative correlation with metres per minute”.

Introduction, line 37: Is it the training load monitoring that may maximise performance or does monitoring help to inform training load management?

RESPONSE: Monitoring actually help to inform training load management. Thank you very much for pointing this out. This part is amended accordingly, and text now reads: “Therefore, proper training load monitoring may help in load management, what consequently may maximise physical performance, reduce occurrences of injury and illness, and minimise the risk of nonfunctional overreaching”. (Please see highlighted text – first paragraph of the Introduction)

Line 52: I would remove the word 'now' as sRPE has been used for a while. Please insert original Foster references here.

RESPONSE: “Now” is excluded as suggested, and original Foster reference is added here. Thank you.

Line 70: Please add "However, the research to date has mostly"

RESPONSE: Amended accordingly. Text now reads “However, the research to date have mostly examined relationships between ETL and ITL by analysing youth or semiprofessional players” (please see highlighted text; 4th line 70-73)

Line 72: Whilst the research on adult elite soccer players may be scarce, please link in relevant studies here on adult elite team sport athletes.

RESPONSE: Amended accordingly. Studies on adult elite players from rugby and Australian football are added, specifically:

  • Lovell, T.W., et al., Factors affecting perception of effort (session rating of perceived exertion) during rugby league training. International Journal of Sports Physiology and Performance, 2013. 8(1): p. 62-69.
  • Gallo, T., et al., Characteristics impacting on session rating of perceived exertion training load in Australian footballers. Journal of Sports Sciences, 2015. 33(5): p. 467-475

Table 1: Please add 2 further rows: intensity within the session and volume per session. That is, the two variables you have used to calculate sRPE.

RESPONSE: Thank you for this comment. We must say that we did not use one specific variable to assess intensity or volume of the sessions. From our practical experience as fitness coaches in elite soccer, both intensity or volume of the sessions cannot be assessed with only one variable. It is typically to assess it with combination of different variables. For example, total distance and total number of accelerations and decelerations can be used to ass volume of the training, while high-intensity running distance, sprint distance, high-intensity accelerations/decelerations are typically used to assess intensity of sessions. Accordingly, this suggestion can not be addressed as asked. If reviewer can provide another detailed instruction to improve table 1, we will be glad to accept it.

Line 125: It would be worth clarifying that the CR-10 Borg Scale is curvilinear (i.e. 5 = hard).

RESPONSE: We agree, thank you for this suggestion. It is now explicitly stated that CR-10 Borg Scale is curvilinear. Text reads: ”Each player’s rating of perceived exertion (RPE) was collected in isolation, 20 minutes after each training session, using the curvilinear CR-10 Borg scale modified by Foster et al.” (please see highlighted text, lines 124-126).

Discussion, page 6, line 197: Please change from 'are' to 'were' as these reflect your direct findings of the research that has been done. That is, adding to the overall body of evidence on TL.

RESPONSE: Amended accordingly. 

Line 228: No need for additional end bracket here.

RESPONSE: Bracket is deleted now. Thank you.

Line 277-278: Can you contrast your analytical methods with non-parametric methods here?

RESPONSE: Thank you for your suggestion. Indeed, usage of the non-parametric variables will hopefully result in more profound interpretation. It is now stated and text reads: “Finally, in this study we observed normally distributed variables only, and consequently parametric statistical methods were applied. Meanwhile, some potentially important non-parametric variables were not used in study, which would allow additional insight into studied relationships.” (please see Limitation section – highlighted text).

References: Please ensure consistent in the capitalisation of journal titles.

RESPONSE: Thank you for noticing this. We reviewed all references according to your suggestion, and now capitalisation of journal titles is consistent.

Line 198: Similarly with 'predict', change to 'predicted'.

RESPONSE: Amended accordingly. 

Thank you once again!

Authors

Reviewer 2 Report

Firstly, I'd like to say I enjoyed reading the study and thought the introduction was particularly well presented and it nicely addresses some simple but important issues around training load monitoring that I feel many practitioners (including some who I work with!) could do well to understand better.  

My first comment would be around the use of sRPE as a measure of internal load. While I can understand the idea of using sRPE as an alternative to true methods of monitoring internal load such as heart rate monitoring, I think there could be more acknowledgement in the introduction of the issues in soccer with regards to collecting accurate and valid data for sRPE within a team sport environment where various factors may impact the values provided by players, not just their perception of training load.

There are a few instances throughout the text where extra not necessary words are used. An example of this is on lines 47 & 48 - 'distances covered in difference speed zones, accelerations, and decelerations using the Global Positioning System (GPS) [8].'

There are also a few instances throughout the text where some small words are missing. An example of this is on line 72 & 73 - 'To the best of our knowledge, only a few studies have investigated this issue to date [27, 30]' 

The above 2 points just need checking through the document. 

In the conclusions, there is a line which states that knowing the external load variables that influence internal training load the most can reduce the risk of injury to players. I think this either needs removing or further justification and reference is required as to how understanding perception of training load is linked to injury (lines 288-291). 

Author Response

Reviewer 1

Firstly, I'd like to say I enjoyed reading the study and thought the introduction was particularly well presented and it nicely addresses some simple but important issues around training load monitoring that I feel many practitioners (including some who I work with!) could do well to understand better. 

RESPONSE: Thank you very much for your kind words. We tried to follow all your suggestions and comments. Please find our responses bellow. 

My first comment would be around the use of sRPE as a measure of internal load. While I can understand the idea of using sRPE as an alternative to true methods of monitoring internal load such as heart rate monitoring, I think there could be more acknowledgement in the introduction of the issues in soccer with regards to collecting accurate and valid data for sRPE within a team sport environment where various factors may impact the values provided by players, not just their perception of training load.

RESPONSE: Thank you very much for this comment. As long-term soccer practitioners (fitness coaches), we totally understand your concerns regarding this issue. Yes, we agree that various factors may impact the values provided by players. While some of them can be controlled (please see Methods where we clearly stated that sRPE was collected in isolation), some of them is impossible to control. According to your comment, we emphasized this issue in Introduction. Text now reads: “Although within a team sport environment various factors may impact the values provided by players, these findings confirm the validity and reliability of this method, leading to widespread use of sRPE for monitoring” (please see end of the second paragraph in Introduction section).

There are a few instances throughout the text where extra not necessary words are used. An example of this is on lines 47 & 48 - 'distances covered in difference speed zones, accelerations, and decelerations using the Global Positioning System (GPS) [8].'

There are also a few instances throughout the text where some small words are missing. An example of this is on line 72 & 73 - 'To the best of our knowledge, only a few studies have investigated this issue to date [27, 30]'

The above 2 points just need checking through the document.

RESPONSE: Thank you for noticing this. These redundant words are now deleted.

Please mind that, as we are not native English speakers, this manuscript has been edited by professional editing service to check quality of language prior to submission. Please find editing certificate here: https://www.dropbox.com/s/3m331gzesj2x1gb/English-Editing-Certificate-47112.pdf?dl=0

However, according to your comment, whole manuscript has been checked one more time.

In the conclusions, there is a line which states that knowing the external load variables that influence internal training load the most can reduce the risk of injury to players. I think this either needs removing or further justification and reference is required as to how understanding perception of training load is linked to injury (lines 288-291).

RESPONSE: Thank you for this suggestion. We agree with you, and we removed this part to avoid possible confusions.

Thank you once again!

Authors